# Influence of the Obtaining Method on the Properties of Amorphous Aluminum Compounds

**Aliya N. Mukhamed'yarova \*, Oksana V. Nesterova, Kirill S. Boretsky, Juliya D. Skibina, Avgustina V. Boretskaya, Svetlana R. Egorova and Alexander A. Lamberov**

Alexander Butlerov Institute of Chemistry, Kazan Federal University, Kazan 420008, Russia;
OVNesterova@kpfu.ru (O.V.N.); KSBoretsky@kpfu.ru (K.S.B.); JDSkibina@stud.kpfu.ru (J.D.S.);
ger-avg@mail.ru (A.V.B.); svetlana.egorova@kpfu.ru (S.R.E.); alexander.lamberov@kpfu.ru (A.A.L.)
\* Correspondence: anm03@list.ru; Tel.: +7-960-051-3178

**Abstract:** Amorphous aluminum compounds are formed during the synthesis of the $\gamma$-Al$_2$O$_3$ catalyst precursor. Amorphous compounds influence on the alumina catalyst variously due to different physicochemical properties, which depend on the method of their preparation. In this research, the comparative analysis of physicochemical properties of amorphous aluminum compounds that were obtained by the precipitation method, the thermal decomposition of aluminum nitrate, and alcoxide hydrolysis product were studied. It is the first time that a new method for calculating of quantitative phase composition of amorphous aluminum compounds using the X-ray powder diffraction, thermogravimetric and differential scanning calorimetry analysis, mass-spectrometry, and CHN-analysis was described. Properties of obtaining samples were studied using scanning electron microscopy, low-temperature nitrogen adsorption, and temperature programmed desorption of ammonium analyses. The methods of precipitation and thermal decomposition of aluminum nitrate allows for obtaining non-porous samples consisting of a mixture of amorphous phases (hydroxide and basic salt) that contain the metals impurities and have low acidity of the oxides obtained from them. The highly porous amorphous alumina formed by the thermal decomposition of the alcoxide hydrolysis product with the least amount of impurities and a high acidity of the surface was observed.

**Keywords:** amorphous aluminum compound; alumina catalyst; alumina; pseudoboehmite; precipitation; thermal decomposition

## 1. Introduction

Gamma-alumina ($\gamma$-Al$_2$O$_3$) is widely used in the petrochemical industry as a catalyst, catalyst carrier, and adsorbent. $\gamma$-Al$_2$O$_3$ is formed during the thermal decomposition of the boehmite [1], obtained from the sol-gel method from bayerite [2], bauxite [3], salts and salt-like aluminum compound [4–6], amorphous aluminum oxide, and hydroxide [5,7]. Depending on the nature and characteristics of the precursor, it is possible to regulate the physicochemical properties of $\gamma$-alumina. One of the most important characteristics of the catalyst precursor is the properties of its surface, such as morphology, porous system, and acidity.

On an industrial scale, $\gamma$-Al$_2$O$_3$ is predominantly produced by the thermal decomposition of pseudoboehmite (PB). There are many methods for the synthesis of PB, for example, a hydrothermal treatment of gibbsite or alumina, a precipitation from solution salts under hydrothermal conditions [8,9], and others. However, the most common methods of PB obtaining are the hydrolysis of aluminum alkoholates [10] and the precipitation of aluminum hydroxide from Al-containing acidic solutions by a base [11]. These methods allow for controlling the phase composition and physicochemical properties of the obtained aluminum hydroxides. In both cases, the formation of

amorphous compounds is observed in addition to the target product due to the forming of incomplete hydrolysis products in the first case and the organic residues in the second case, in accordance with Scheme 1.

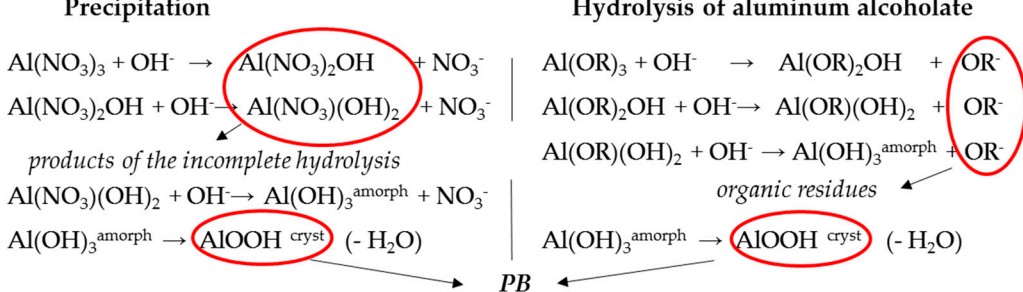

**Scheme 1.** Forming of X-ray amorphous aluminum compounds by the precipitation and hydrolysis.

Currently, the amorphous aluminum compounds have been studied poorly and they are described in a small amount of research works. However, these compounds affect the properties of $\gamma$-$Al_2O_3$ catalysts.

On the one hand, the amorphous component presence as a part of alumina catalyst precursor in various petrochemical processes is unfavorable. The amorphous compound is a gel that reduced the filterability of the precipitation product and complicated the washing of PB from impurity ions (Na, Fe, etc.) that are catalytic poisons. In addition, the amorphous component has a low specific surface area [12]. Therefore, the amount of the amorphous component in the PB is reduced at the synthesis process.

On the other hand, the amorphous component has a positive effect on the catalyst. An amorphous base aluminum salts provide the plastic properties of the catalyst precursor during a molding. The salts decompose with the formation a new binding phase after the calcination [13]. In our previous works [14], we have already shown that it is possible to improve the properties of catalyst for skeletal isomerization of n-butenes to isobutylene by transforming the amorphous component using hydrothermal treatment.

There are several ways to obtain the amorphous aluminum compounds: a calcination of the aluminum salts [7], the hydrolysis of the aluminum alcoholates [15], and the precipitation from Al-containing solutions by the base [7,16]. As a result, various non-crystalline aluminum compounds are formed differing in a phase composition and the physicochemical properties of surface. High-purity amorphous aluminum hydroxide containing the organic residues is obtained by the hydrolysis of organoaluminum compound. The residues of salts precursor are also presented in the structure of amorphous compounds that are formed by the precipitation and the thermal decomposition of aluminum salts, for example, nitrate, according to the Scheme 2.

**Thermal decomposition**

$$[Al(H_2O)_6(NO_3)_3] \cdot 3H_2O \xrightarrow{80\,°C} Al(H_2O)_6(NO_3)_3 \, (-H_2O)$$

$$Al(H_2O)_6(NO_3)_3 \xrightarrow{>170\,°C} Al(H_2O)_x(NO_3)_y \, (-NO-H_2O)$$

*products of incomplete thermal decomposition*

**Scheme 2.** Forming of X-ray amorphous aluminum compound by the thermal decomposition.

These schemes are debatable, and further research is needed to clarify them.

Therefore, the purpose of our research was a comparative analysis of physicochemical surface properties of the amorphous aluminum compounds that were obtained by the methods, such as the

precipitation method, the thermal decomposition of aluminum nitrate, and alcoxide hydrolysis product. We also propose a new method for calculating the phase composition of the amorphous aluminum compounds using a group of XPRD, TG-DSC, mass spectrometry (MS), and CHN-analysis methods.

## 2. Materials and Methods

Three samples of the amorphous aluminum compounds were obtained by the various methods. Sample 1 (S1) was carried out by the precipitation from 0.25 g/mL $Al(NO_3)_3 \cdot 9H_2O$ (analytically pure, according to the State standard GOST 3757-75 [17] contains <0.004 wt % Fe, <0.05 wt % K + Na) solution by ammonia water (chemically pure, according to the State standard GOST 3760-79 [17] contains <0.0001 wt % Fe, <0.0001 wt % Mg + Ca, <0.001 wt % $CO_3^{2-}$) at pH = 6.0 and the room temperature without the stabilization and aging stages. The obtained gel-like product was centrifuged 20 min in 700 mL of $H_2O$ 4 times to remove the ammonium nitrate formed during the precipitation. Subsequently, sample S1 was drying during 2 h at the 100–105 °C. A precipitate with the maximum amount of basic salt was obtained. Sample 2 (S2) was synthesized by heat treatment of $Al(NO_3)_3 \cdot 9H_2O$ (analytically pure, the State standard GOST 3757-75) at 350 °C during 1 h at an atmospheric pressure. Sample 3 (S3) was obtained by heat treatment at 550 °C during 2 h at the atmospheric pressure from non-crystalline aluminum hydroxide (impurities: 0.52 wt % C, <0.001 wt % Na, Mg and Fe) that was obtained by the method of hydrolysis of aluminum isopropoxide (chemically pure, <0.0001 wt % Na, Mg, Fe) by water vapor. The hydrolysis of aluminum isopropoxide with water vapor was carried out by the following equation: $Al(i\text{-}C_3H_7O)_3{}^{\text{liquid}} + 3H_2O{}^{\text{vapor}} = Al(OH)_3{}^{\text{solid}} + 3C_3H_7OH{}^{\text{vapor}}$

The resulting aluminum hydroxide ($Al(OH)_3{}^{\text{solid}}$) (Figure S1 in the Supplementary Material) is an amorphous compound with organic residues, which was heat treated at 550 °C/2 h up to their complete removal.

The element composition of the samples was made using a system for the CHNS/O analysis of PE 2400-II (Perkin Elmer, Waltham, MA, USA) and an iCAP Q inductively coupled plasma mass spectrometer (ICP-MS) (Thermo Fisher Scientific, Waltham, MA, USA).

The phase composition of samples was determined using X-ray powder diffraction (XRPD) by a MiniFlex 600 diffractometer (Rigaku, Tokyo, Japan) equipped with a D/teX Ultra detector. In this experiment, Cu K$\alpha$ radiation (40 kV, 15 mA) was used and data was collected at room temperature in the range of 2θ from 2° to 100° with a step of 0.02° and exposure time at each point of 0.24 s without sample rotation. The phase concentrations were determined using the thermal analysis (TA; Netzsch STA–449C Jupiter, Selb, Germany). The TA was carried out in a way that is capable of recording the thermogravimetric (TG), derivative thermogravimetric (DTG), and differential thermal analysis (DTA) curves simultaneously. The samples were heated in the temperature range of 30–1000 °C at the uniform heating rate of 10 °C/min in an argon flow. Concentrations of aluminum hydroxides were calculated from the amount of the water that was released during the aluminum hydroxides dehydration and dehydroxilation. Mass spectrometry (MS) analysis was carried out at the heating of samples at the same rate in He current using of a ThermoStar GSD 320 T gas analyzer (Pfeiffer Vacuum, Nashua, NH, USA).

The measurements by scanning electron microscopy (SEM) were carried out with an EVO 50XVP electron microscope combined with an INCA 350 energy-dispersive spectrometer (Carl Zeiss, Upper Cohen, Germany). The spectrometer resolution was 130 eV. The analysis was carried out at an acceleration voltage of 20 kV and flange back distance of 8 mm.

The specific surface area $S_{BET}$ (calculated with the Brunauer–Emmett–Teller method [18]) and the pore volume $V$ (calculated by the last point of isotherm [18]) were determined using a multipurpose Autosorb-iQ analyser (Quantachrome Instruments, Boynton Beach, FL, USA). Adsorption isotherms were obtained at −196 °C (77 K) after the degassing of samples at 150 °C under the residual pressure of 0.013 Pa. The pore volumes' distributions over the pore diameters were calculated from the curve of desorption isotherm by using the standard Barrett–Joyner–Halenda mechanism ($V_{BJH}$) [18].

The surface acidity of S3 sample and alumina derived from the S1 and S2 samples was analyzed by the method of temperature-programmed desorption of ammonia (TPD-NH$_3$) on a flow-type instrument with a thermal conductivity detector ChemBet Pulsar (Quantachrome Instruments, Boynton Beach, FL, USA). Before ammonia adsorption, the sample was degassed by heating up to 550 °C in a helium flow. The adsorption step was carried out in a stream of ammonia for 30 min at a temperature of 100 °C. The physically adsorbed ammonia was removed by helium flow at 100 °C for 30 min. After analysis, the sample was cooled to room temperature in a helium flow. Temperature programmed desorption of ammonia was carried out from room temperature to 700 °C at a rate of 10 °C/min. Calculations of TPD-NH$_3$ data on the distribution of acid sites were performed according to the method that is given in the research [19].

## 3. Results and Discussion

All the samples of amorphous compounds obtained by different methods contain 95–99 wt % of the basic elements of Al, H, O, and differ from the impurity composition (Table 1). The most amount of impurities is identified in sample S2 obtained by heat treatment of aluminum nitrate Al(NO$_3$)$_3$·9H$_2$O, which consisted of the nitrate according to the State standard (GOST 3757-75). Sample S3 was synthesized from the high-purity product of the hydrolysis of aluminum alcoxide. Therefore, after calcining, S3 contains the minimum amount of Li, C, N, Na, and Mg impurities. Samples S1 and S2 include a considerable amount of nitrogen in the form of NO$_3^{3-}$ group (Table 1). S1 is the product by the precipitation from the solution of aluminum nitrate with ammonia where a nitrogen in its structure remains due to incomplete hydrolysis. S2 is the product of thermal decomposition of aluminum nitrate, the heat treatment leads to the incomplete release of nitrogen oxides from the nitrate when residual water molecules are still present [6].

**Table 1.** Impurity content of samples S1–S3.

| Sample | Obtaining Method | Conditions | Content (wt %) | | | | | |
|--------|------------------|------------|------|------|------|------|------|------|
| | | | Na | Mg | Li | C | N | Fe |
| S1 | Precipitation | pH = 6.00 | <0.009 | <0.001 | 0.000 | 0.41 | 4.57 | <0.009 |
| S2 | Heat treatment of aluminum nitrate | 350 °C, 1 h | 0.017 | 0.015 | 0.005 | 0.21 | 1.71 | 0.011 |
| S3 | Heat treatment of hydrolysis product | 550 °C, 2 h | 0.007 | 0.011 | 0.000 | ~0.05 | 0.00 | 0.022 |

Samples S1–S3 are identified on the diffractograms as the X-ray amorphous aluminum compounds. A wide halo effect at the range of 20°–40° of the S2 and S3 and of S1 are shown in Figure 1. The diffractogram of S1 is characterized by the diffuse diffraction lines of PB at 20°–70° with $d/I_{(120)}$ = 0.316 nm, $d/I_{(031)}$ = 0.235 nm, $d/I_{(200)}$ = 0.177 nm, $d/I_{(002)}$ = 0.143 nm (JCPDS Card 00-021-1307), and a diffraction line at 8° with the $d/I_{(100)}$ = 1.203 nm of the aluminum complex Al$_x$(OH)$_y$(NO$_3$)$_z$(NH$_4$) (JCPDS Card 01-080-7534) (Figure 1). The presence of PB of the sample S1 is also confirmed by TG-DSC and MS analyses. According to the TG-DSC data, PB is exhibited by a shoulder of the II endothermic effect on the DSC curve of sample S1 at the temperature of 397–570 °C with a mass loss ($\Delta m$) of 3.63 wt %, which is accompanied by the release of H$_2$O and NO (Figure 2a,b):

$$Al_2O_3 \cdot nH_2O \xrightarrow{400-550 \text{ °C}} \gamma - Al_2O_3 + nH_2O \uparrow$$

On the DSC curves of all samples, two endothermic effects and the exothermic effect (Figure 2a–f, Table 2) are observed due to the following transformations [20,21]:

$$\text{I endothermic effect}: \ Al_2O_3 \cdot 3H_2O^{\text{amorph}} \xrightarrow{200-300 \text{ °C}} Al_2O_3{}^{\text{amorph}} + 3H_2O \uparrow$$

II endothermic effect : $Al_2O_3 \cdot xH_2O \cdot yNO^{amorph} \xrightarrow{300-400\,°C} Al_2O_3^{amorph} + xH_2O\uparrow + yNO\uparrow$

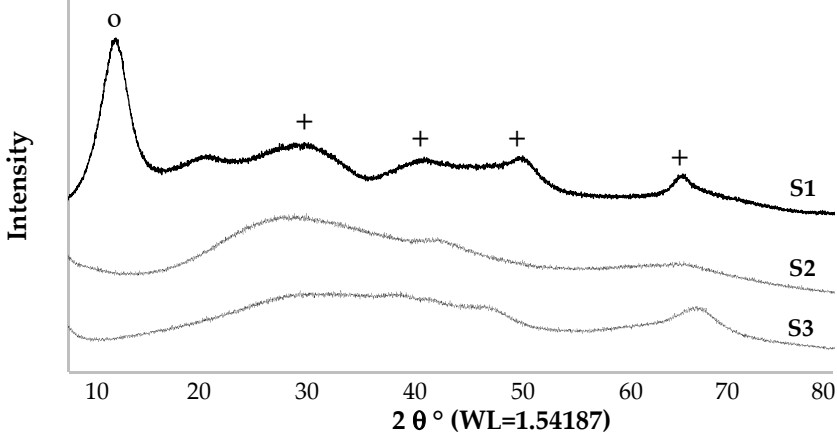

**Figure 1.** Diffractograms of samples S1–S3 (+, PB; o, aluminum complex).

The I endothermic effect corresponds to the release of water from the porous system of samples at the temperature up to 200 °C and during thermal decomposition of amorphous aluminum hydroxide $Al_2O_3 \cdot xH_2O^{\ amorph}$ above 200 °C. The II endothermic effect at the 321–495 °C is identified as the thermal decomposition of the basic aluminum salts $Al_2O_3 \cdot xH_2O \cdot yNO^{\ amorph}$ due to the release of $H_2O$ and NO (Table 2, Figure 2a–d), which corresponds to $m/z$=18 and 30, respectively, on the mass spectra.

The exothermic effect on the DSC curves is due to phase transformation low-temperature $Al_2O_3$ to high-temperature oxide without changing the mass (Figure 2a–f, Table 2), according to the following scheme [20]: $Al_2O_3^{amorph} \rightarrow \gamma\text{-}Al_2O_3 \rightarrow \theta\text{-}Al_2O_3 \rightarrow \alpha\text{-}Al_2O_3$. The amorphous alumina ($Al_2O_3^{amorph}$) is formed due to thermal decomposition of X-ray amorphous aluminum hydroxide and basic aluminum salts [7]. The obtained from PB $\gamma\text{-}Al_2O_3$ is transformed to high-temperature alumina without a visible exothermic effect on the S1 DSC curve [1,4,7,12,22] (Figure 2a,b).

**Table 2.** TG-DSC and mass spectrometry (MS) analyses results of samples S1–S3.

| Sample | Obtaining Method | Conditions | Endothermic Effects | | | | Exothermic Effect | |
|---|---|---|---|---|---|---|---|---|
| | | | I Effect | | II Effect | | | |
| | | | $T_p$ $(T_b-T_f{}^*)$ (°C) | $\Delta m$ (wt %) | $T_p$ $(T_b-T_f)$ (°C) | $\Delta m$ (wt %) | $T_p$, °C | $T_b-T_f$, °C |
| S1 | Precipitation | pH = 6.00 $T$ = 25 °C | 258 (104–276) | 14.59 | 354 (321–397) | 23.48 | 863 | 814–907 |
| S2 | Heat treatment of aluminum nitrate | 350 °C, 1 h | 198 (129–278) | 1.53 | 399 (339–495) | 8.89 | 884 | 855–911 |
| S3 | Heat treatment of hydrolysis product | 550 °C, 2 h | 102 (30–140) | 10.69 | – | – | 799 | 683–882 |

\* $T_p$, peak temperature; $T_b$, temperature of effect beginning; $T_f$, temperature of effect finish.

Hereby, the method of the precipitation from a solution of aluminum nitrate with ammonia at pH = 6.0 without stabilization and aging (sample S1) allows for obtaining a complex amorphous compound in the form of a phase mixture that consists of amorphous aluminum hydroxide, basic aluminum salt, and poorly crystallized pseudoboehmite.

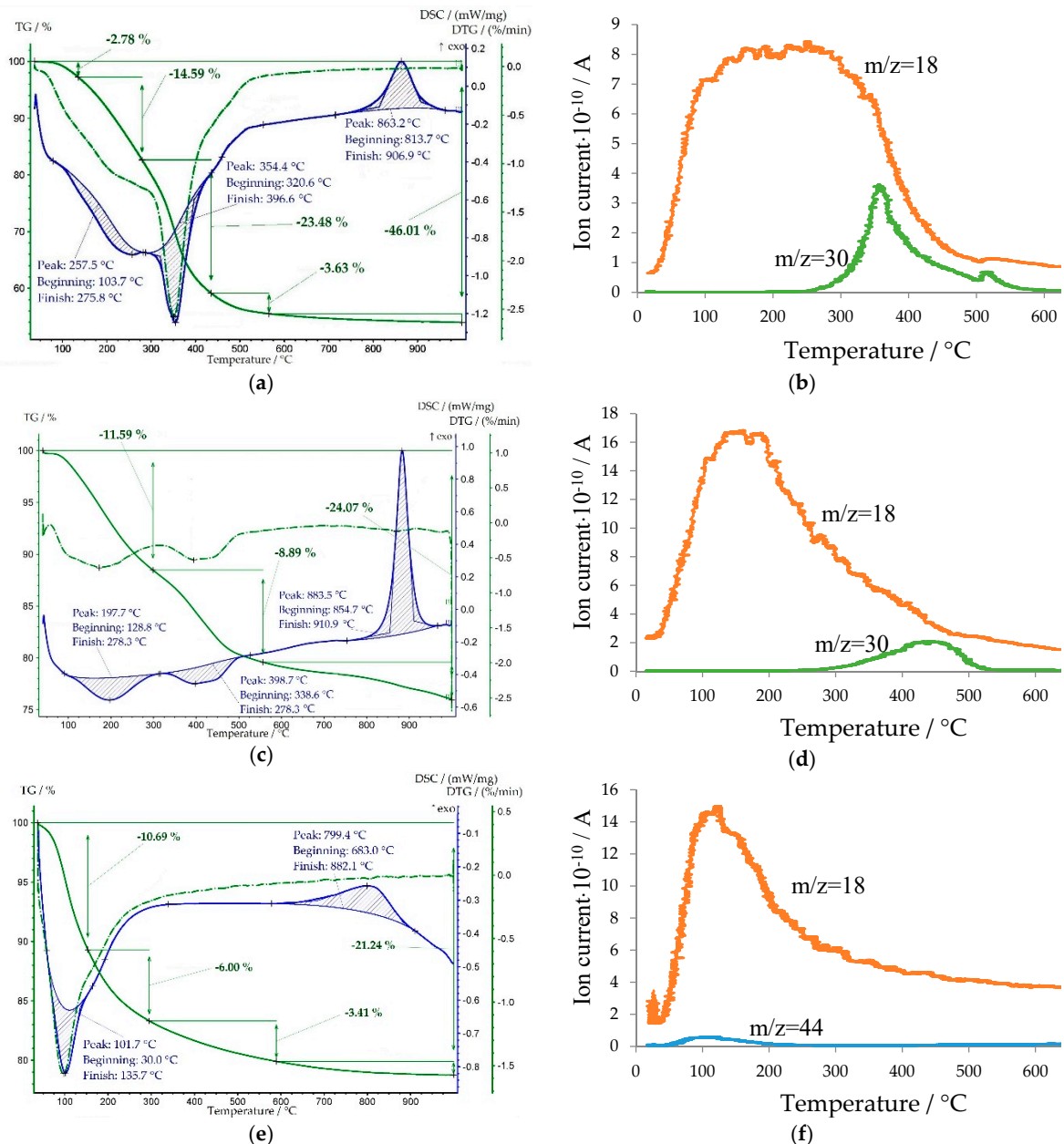

**Figure 2.** TG-DSC and MS analyses results of samples: S1 (**a**,**b**), S2 (**c**,**d**), S3 (**e**,**f**).

The quantitative phase composition of the complex precipitation product (S1) was calculated from the mass loss of the corresponding endothermic effects on the DSC curve and the areas of the NO peak in the mass spectrum (Figure 2a,b), since other nitrogen oxides are almost not observed. Based on the total nitrogen content in the sample (Table 1), the total amount of NO released from S1 during thermal decomposition to 650 °C is 9.79 wt %. NO is exhibited by two peaks on S1 mass spectrum in the temperature range of II endothermic effect and its shoulder on the DSC curve of the sample in a ratio of 28.6:1.0 (Figure 2a,b). In this way, almost all nitrogen is released from the basic aluminum salt structure (only II endothermic effect). There is a possibility that the release of NO 0.32 wt % during thermal decomposition of PB is identified by the shoulder on DSC curve is due to the rearrangement of the hydroxide lattice to oxide (PB $\rightarrow$ $\gamma$-$Al_2O_3$). 3.63 wt % $H_2O$ is also released at the temperature. Since PB in the sample is poorly crystallized, as evidenced by diffuse diffraction lines (Figure 1), the value of interlayer water *n* in its structure was taken as the maximum, i.e., *n* = 2.0 [23]. In this case, the amount of PB in the sample S1 is 12.69 wt %.

Based on the total content of NO, 9.47 wt % NO and 14.01 wt % $H_2O$ are released during thermal decomposition of the basic aluminum salt at the II endothermic effect, i.e., 0.32 and 0.78 mol, respectively. Therefore, the empirical salt formula is $Al_2O_3 \cdot 0.78H_2O \cdot 0.32NO$.

It was reported in the following scientific works [7,21,22] that the empirical formula amorphous aluminum hydroxide that was obtained by the precipitation of the aluminum salts by the base is $Al_2O_3 \cdot 3H_2O$. Its thermal decomposition begins at the temperature more than 200 °C. Hence, the mass loss of I endothermic effect on the S1 DSC curve was divided into two parts, 8.18 wt % $H_2O$, of which is physically adsorbed water (<200 °C) and 6.41 wt % $H_2O$ releases during the thermal decomposition of amorphous hydroxide. The calculated amount of basic aluminum salt and amorphous $Al_2O_3 \cdot 3H_2O$ in the sample S1 are 18.52 wt % and 60.61 wt %, respectively.

According to the results of element, XRPD, TG-DSC, and MS analyses, the phase composition of the precipitation product (S1) calculated without taking into account physically adsorbed water is presented in Table 3.

**Table 3.** The phase composition of amorphous S1–S3 compounds.

| Sample | Obtaining Method | Conditions | Empirical Formula | Amount, wt % |
|---|---|---|---|---|
| S1 | Precipitation | pH = 6.00, $T$ = 25 °C | $Al_2O_3 \cdot 3H_2O$ (amorphous hydroxide)<br>$Al_2O_3 \cdot 0.78H_2O \cdot 0.32NO$ (basic salt)<br>$Al_2O_3 \cdot 2H_2O$ (PB) | 20.2<br>66.0<br>13.8 |
| S2 | Heat treatment of aluminum nitrate | 350 °C, 1 h | $Al_2O_3 \cdot 3H_2O$ (amorphous hydroxide)<br>$Al_2O_3 \cdot 0,29H_2O \cdot 0,12NO$ (basic salt) | 16.3<br>83.7 |
| S3 | Heat treatment of hydrolysis product | 550 °C, 2 h | $Al_2O_3$ (amorphous oxide) | 100.0 |

The method of thermal decomposition of aluminum nitrate $Al(NO_3)_3 \cdot 9H_2O$ at 350 °C for 1 h allows for obtaining a complex X-ray amorphous compound in the form of the phase mixture of aluminum hydroxide and basic salt in the amount of, respectively, 16.3 and 83.7 wt % (Table 3). These phases are exhibited by two corresponding endothermic effects on the S2 DSC curve and the release of $H_2O$ and NO on the MS spectrum (Table 2, Figure 2c,d).

The quantitative S2 composition was calculated, as in case of Sample S1, by the mass loss of endothermic effects on the S2 DSC curve (Table 2, Figure 2c,d). Based on the total nitrogen content in the sample (Table 1), the total amount of released NO is 3.66 wt %. The amount of amorphous aluminum hydroxide ($Al_2O_3 \cdot 3H_2O$) was calculated based on mass loss on the S2 DSC curve more than 200 °C and equaled 16.27 wt %. Based on the total content of NO (3.66 wt %) and the mass spectrum of sample S2 (Figure 2d), the release of NO occurs only during the thermal decomposition of salt (II endothermic effect) and 5.23 wt % $H_2O$ is also released, i.e., respectively 0.12 and 0.29 mol. Consequently, the empirical formula of the basic aluminum salt is $Al_2O_3 \cdot 0.29H_2O \cdot 0.12NO$.

The thermal decomposition method of the product of aluminum alcoxide hydrolysis at 550 °C for 2 h (sample S3) allows for obtaining a phase-homogeneous X-ray amorphous alumina, which has a small amount of metal impurities (Table 1). Residual organic compounds are exhibited by a pronounced shoulder of the I endothermic effect in the low-temperature region on the S3 DSC curve and the presence in the mass spectrum of peaks, corresponding to the release of $CO_2$ and $H_2O$ (Figure 2e,f).

The precipitation method (pH = 6.0) allows for obtaining the mixture of particles (~100 μm), the surface of which consists of thin plates of size to 1 μm layering on each other closely, with extended cracks resulting from drying the gel-like precipitate [7,21] (Figure 3a,b). It is impossible to identify the porous system of S1 in the SEM images. Indeed, according to the data of low-temperature nitrogen adsorption, sample S1 has a specific surface area <1 $m^2$/g and a pore volume <0.01 $cm^3$/g (Table 4).

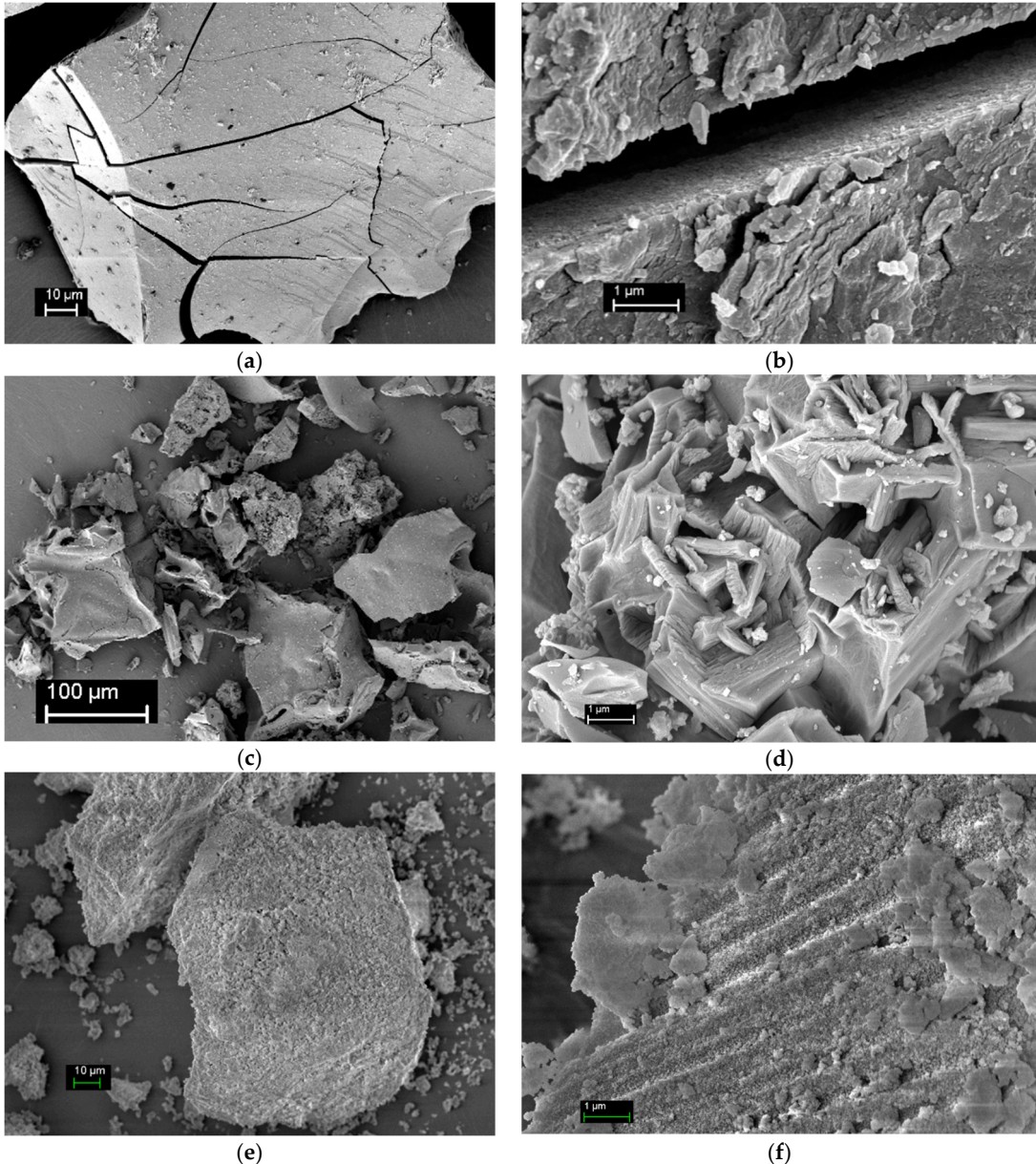

**Figure 3.** Scanning electron microscopy (SEM) images of external and internal surface of samples S1 (**a**,**b**), S2 (**c**,**d**), S3 (**e**,**f**).

The heat-treated product of $Al(NO_3)_3 \cdot 9H_2O$ (sample S2) is a mesoporous compound (as shown in the Figure S2 in the Supplementary Material), which has S and V of 29 $m^2$/g and 0.04 $cm^3$/g, respectively (Table 4, Figure 4a). S2 particles are irregularly shaped particles of > 100 μm (Figure 3c). The external surface is smooth, it has a size of 50–100 μm, and it contains small fragments with cavities. The internal surface of S2 particles includes large cavities in the form of hemispheres with 5–10 μm diameters, sponge-like structure with cavity sizes of 50–850 nm, and it accretes plate-like particles between which conical spaces with a base diameter less than 530 nm are formed (Figure 3d). As shown on the SEM images (Figure 3c,d), sample S2 has the cavities formed, probably at the intense releasing of water and nitrogen oxide molecules during the thermal decomposition of nitrate. The cavities are distributed throughout the sample. Therefore, almost uniform distribution of $V_{BJH}$ pore diameters is observed (Table 4).

Heat treatment at 550 °C of the hydrolysis product makes it possible to obtain a highly porous sample with a high acidity of the surface (Table 2; Table 4, Figure 4b). Sample S3 particles are the

large particles of 3–133 μm, which consist of aggregates of irregularly shaped plates (Figure 3e). The aggregates are formed by small particles of 10–30 nm. There are narrow and large cavities, respectively, of <9 nm and 0.3–2.5 μm between them.

**Table 4.** Porous system parameters of amorphous compounds S1–S3.

| Sample | Obtaining Method | Conditions | $S$, $(m^2/g)$ | $V$, $(cm^3/g)$ | $V_{BJH}$, $(cm^3/g)$ | Distribution of $V_{BJH}$ over $D_p$, $cm^3/g$ (%) | | | $D_{max}$, (nm) $dV/dD$, $(cm^3/g·nm)$ |
|--------|------------------|------------|------|---------|-----------|--------|---------|--------|-------|
| | | | | | | <3 nm | 3–10 nm | >10 nm | |
| S1 | Precipitation | pH = 6.00, $T$ = 25 °C | <1 | <0.01 | <0.01 | – | – | – | – |
| S2 | Heat treatment of aluminum nitrate | 350 °C, 1 h | 29 | 0.04 | 0.05 | 0.02 (40) | 0.02 (40) | 0.01 (20) | 3.7/0.02 |
| S3 | Heat treatment of hydrolysis product | 550 °C, 2 h | 344 | 0.90 | 0.95 | 0.01 (1) | 0.56 (59) | 0.38 (40) | 4.5/0.19 |

In the crack of aggregate, parallel channels with a diameter of 280–360 nm are formed (Figure 3f), the entrance to which is the larger wells of ~400 nm diameter on the external surface of particle. Despite the absence of micropores, sample S3 has high S (344 m²/g), which includes 59% pores with $D_p$ = 3–10 nm (as shown in the Figure S2 in the Supplementary Material and Table 4). Probably, the narrow pores with $D_p$ = 4.5 nm of the sample are the spaces between the primary particles of alumina. The cavities are pores with $D_p$ > 10 nm, which consist of 40 % of the S3 porous system.

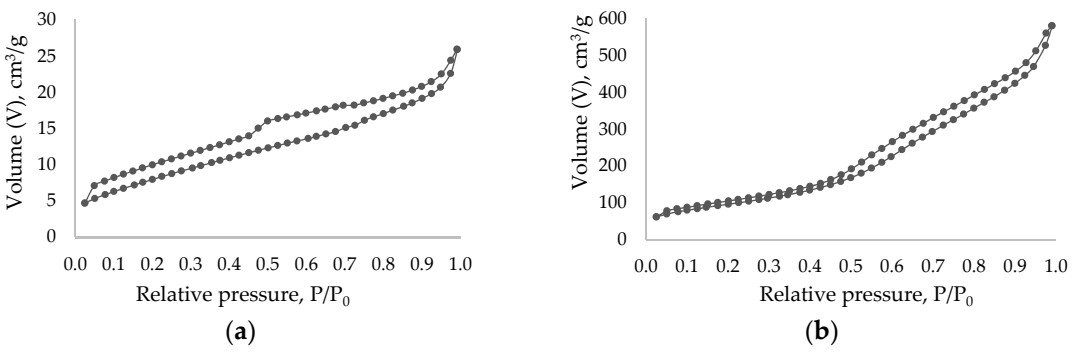

**Figure 4.** Isotherms of samples S2 (**a**) and S3 (**b**).

Acid and base sites possessing various structure and strength are presented on the surface of alumina obtained by the heat treatment (HT) of the amorphous aluminum compounds. The distribution of sites is mainly caused by the phase composition of a compound. Acid sites are Brønsted-type (terminal or bridge surface hydroxyl groups) and Lewis (coordination-unsaturated aluminum cations) centers [24,25]. Both types of acid sites are identified by various methods one of which is the TPD of $NH_3$ method. It allows for determining the total acidity of the surface. However, the proton affinity energy of the strongest Brønsted sites of alumina is not sufficient for the protonation of ammonia molecules [26].

After dehydration, the Lewis acid and basic sites are simultaneously formed on the surface of the alumina. Lewis acid sites transform into Brønsted sites in the presence of water molecules [27]:

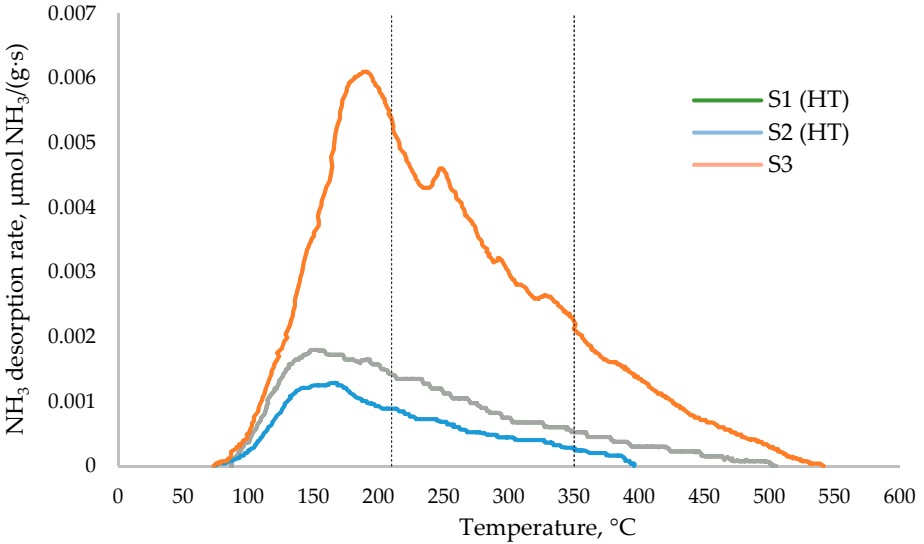

Interacting the $NH_3$ molecule with Lewis acid sites forms a coordination (donor-acceptor) bond due to the free electron pair of the nitrogen atom:

$$\equiv Al \dots NH_3$$

For studying the samples' acid properties, S1 and S2 were previously heat treated (HT) at the 550 °C to remove excess water in the form of hydroxyl groups from the surfaces. Sample S3 was not additionally treated because it was obtained by heating at 550 °C. The TPD curves of $NH_3$ samples are presented in Figure 5. The distribution of acid sites (a.s.) according to the ammonia desorption energy ($E_d$) for the obtained samples is given in Table 5. The a.s. with $E_d$ <110 kJ/mol, which corresponds to the desorption temperature up to 210 °C, were attributed to the weak strength sites. The a.s. with $E_d$ to 110 to 142 kJ/mol and $E_d$ >142 kJ/mol are, respectively, taken for the medium (desorption temperature of 210–350 °C) and the strong strength sites (desorption temperature is more than 350 °C).

**Figure 5.** Curves of the temperature-programmed desorption (TPD) of ammonia of samples (HT-heat treated).

TPD-$NH_3$ curves of all samples begin from the temperature of ~75 °C (Figure 5). Diffuse peaks are noted to the dependences of the ammonia desorption rate on the temperature of S1$^{HT}$ and S2$^{HT}$ samples in the region of ~140–190 °C. An intensive peak is noted on the curve of S3. These peaks are characterized by weak a.s. In the area of medium a.s. of S1$^{HT}$ and S2$^{HT}$ samples, the ammonia desorption rate with increasing temperature decreases slowly, while the desorption rate on the S3 curve increases with a peak at the temperature of 235–250 °C. In the field of strong a.s., an equable decrease in the desorption rate to 400, 500, and 540 °C occurs for samples S1$^{HT}$, S2$^{HT}$, and S3, respectively.

**Table 5.** Ammonia ($NH_3$) TPD acid characteristics of samples S1$^{HT}$, S2$^{HT}$, and S3.

| Sample | Obtaining Method | Conditions | Number of a.s., μmol $NH_3$/g | | | |
|---|---|---|---|---|---|---|
| | | | $E_d < 110$, (kJ/mol) | $110 < E_d <$ 142, (kJ/mol) | $E_d > 142$, (kJ/mol) | Total |
| S1$^{HT}$ | Precipitation | pH = 6.00, $T$ = 25 °C | 99.1 | 77.8 | 23.3 | 200.2 |
| S2$^{HT}$ | Heat treatment of aluminum nitrate | 350 °C, 1 h | 45.6 | 63.2 | 7.1 | 115.9 |
| S3 | Heat treatment of hydrolysis product | 550 °C, 2 h | 150.9 | 427.6 | 73.7 | 652.2 |

In the case of a precipitation product S1$^{HT}$, the intermediate value of acidity in comparison with S2$^{HT}$ and S3 samples is due to the content of Li, Na, Mg and Fe impurity ions (Table 1). Also, after the heat treatment of basic aluminum salts, a small amount of a.s. on the surface of the amorphous oxide is probably due to the formation of the base sites during the removing of NO and $H_2O$ molecules to the following scheme:

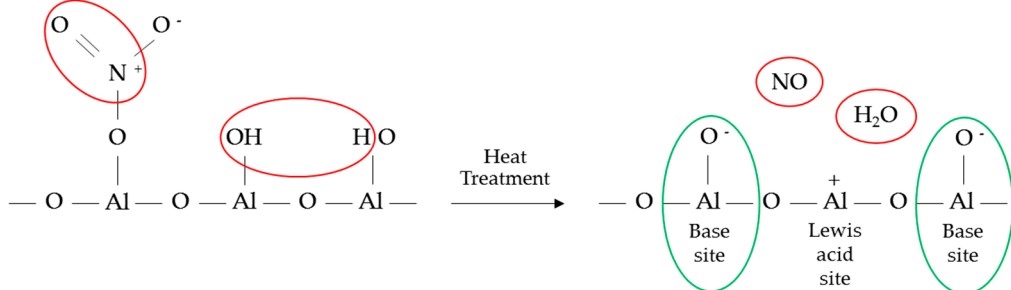

$\gamma$-$Al_2O_3$ also contributes to the total a.s. quantity [25]. Indeed, S1$^{HT}$ contains the strong a.s. and medium a.s. in amounts of 12% and 38%, respectively (Table 5).

After heat treatment of the sample S2, amorphous hydroxide and basic aluminum salt form amorphous alumina that possesses a small amount of a.s., 96% of which are weak and medium strength (Table 5, Figure 5). As previously noted, this effect is observed due to the high content of Na, Mg, and Fe impurity ions (Table 1). Also, the small amount of a.s. is probably due to the formation of base sites during the removal of NO and $H_2O$ molecules after the thermal decomposition of nitrogen-containing basic aluminum salts.

Amorphous alumina obtained by thermal decomposition of the alcoxide hydrolysis product has a large number of a.s. on the surface, 65% of which is medium strength (Table 4). S3 consists of the small amount of Na (Table 1) and the impurities do not block a.s. its surface as against the S1$^{HT}$ and S2$^{HT}$. In addition, during the heat treatment of the hydrolysis product, thermal decomposition of underhydrolyzed alcoholate occurs with the formation of $CO_2$ molecules, the O$^-$ atom of which was probably bended with Al. Hence, the Lewis a.s. is formed according to the following scheme:

## 4. Conclusions

The comparative analysis of amorphous aluminum compounds obtained by the precipitation method, the thermal decomposition of aluminum nitrate and alcoxide hydrolysis product, and their physicochemical properties were studied. For the first time a new method for calculating quantitative phase composition of amorphous aluminum compounds using a group of methods as XPRD, TG-DSC, MS, and CHN-analysis was described.

The method of precipitation of 0.25 g/mL of aluminum nitrate by ammonia solution at pH = 6.0 without aging and stabilization stages allows for obtaining a complex amorphous aluminum compound, which includes amorphous hydroxide ($Al_2O_3 \cdot 3H_2O$), basic salt ($Al_2O_3 \cdot 0.78H_2O \cdot 0.32NO$), and pseudoboehmite ($Al_2O_3 \cdot 2H_2O$). This compound is non-porous, its particles has large cracks. For the heat-treated precipitation product, which is a mixture of amorphous alumina and $\gamma$-alumina, is characterized by the total number of a.s. of 200.2 $\mu$mol/g containing 12% strong and 50% weak acid sites on its surface due to the presence in the sample of Li, Na, Mg, and Fe impurity ions.

The method of thermal decomposition of aluminum nitrate at 350 °C for 1 h allows for obtaining a complex amorphous aluminum compound, which includes amorphous hydroxide ($Al_2O_3 \cdot 3H_2O$) and basic salt ($Al_2O_3 \cdot 0.29H_2O \cdot 0.12NO$). This compound is mesoporous with the low specific surface area and pore volume, a maximum on the differential distribution curve of pore volumes over the pore diameter is 3.7 nm. The product of thermal decomposition of nitrate is a monolithic irregularly shaped particle with cavities that were evenly distributed throughout the sample. The thermal decomposition of the sample leads to amorphous oxide, which is accompanied by the formation of a small quantity of acid sites on its surface, which contains 6% strong and 55% medium sites due to the high content of Na, Mg, and Fe impurities in the sample.

The method of heat treatment at 550 °C for 2 h of the hydrolysis product of the alcoxide allows for obtaining a monophasic amorphous alumina. It is highly porous with high specific surface area and pore volume, a maximum on the distribution curve of pore volumes over diameters is 4.5 nm. Amorphous alumina is a large aggregates formed by small particles, between which are narrow and large cavities. It has a large quantity of acid sites on the surface, which is 65% medium and 12% strong sites due to the small amount of impurities in the sample.

**Supplementary Materials:** The following are available online at http://www.mdpi.com/2079-6412/9/1/41/s1, Figure S1. Diffractogram and thermal curves of amorphous aluminum hydroxide obtained by the hydrolysis (S3 sample precursor); Figure S2. The curves of distribution of $V_{BJH}$ with $D_p$ of samples S2 (a) and S3 (b).

**Author Contributions:** Conceptualization, A.N.M. and S.R.E.; Methodology, S.R.E. and A.A.L.; Validation, A.N.M., S.R.E. and A.A.L.; Formal Analysis, O.V.N., K.S.B., A.V.B. and J.S.; Investigation, A.N.M. and A.V.B.; Resources, A.A.L.; Data Curation, A.N.M.; Writing-Original Draft Preparation, A.N.M.; Writing-Review & Editing, A.N.M.; Visualization, A.N.M.; Supervision, S.R.E. and A.A.L.; Project Administration, A.N.M. and S.R.E.; Funding Acquisition, A.N.M. and A.A.L.

**Funding:** The reported study of samples S1 and S2 (products of precipitation aluminum nitrate by ammonia solution and thermal decomposition of aluminum nitrate) was funded by RFBR according to the research project No. 18-33-00559. The study of sample S3 (product of thermal decomposition of aluminum hydroxide obtained by the organoaluminum compounds hydrolysis) was performed according to the Russian Government Program of Competitive Growth of Kazan Federal University.

**Acknowledgments:** The thermal analysis of samples was carried out at the Federal Center for Collective Use of the Kazan Federal University with the support of the Russian Agency for Science and Innovation A.V. Gerasimov. The scanning electron microscopy measurements were performed at the Interdisciplinary Center "Analytical Microscopy" of the Kazan Federal University V.V. Vorobiev and V.G. Evtyugin respectively.

**Conflicts of Interest:** The authors declare no conflict of interest.

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
