# Peer review of "Influence of the Obtaining Method on the Properties of Amorphous Aluminum Compounds"

_coatings, doi:10.3390/coatings9010041_

Round 1

Reviewer 1 Report

The paper describes the characterization of 3 X-ray amorphous alumina samples. Overall, the English and especially the title should be improved. The authors should take care to use exact scientific terms.

Comments:

Line 16 : alcoholates hydrolysis product: the usual term is “alcoxyde”, alcoholates is more generic

Line 81 : MS as standing alone technique? do you mean ICP-MS?, ATG coupled MS?

Line 97: what is Q in the equation?

Line 126: What is the standard “Barret-Joyner –Highland mechanism”? Usually BJH refers to the method of Barrett, Joyner and Halenda, a procedure for calculating pore size distributions from experimental isotherms using the Kelvin model of pore filling. Please describe this mechanism and add literature reference.

Lines 247, 257, 258, 351: crystals? The term crystal is misused in the paper; with 100µm crystals you should have nice XRD patterns….

Lines 253-254: “is a mesoporous compound with meso and micropores”: How can you prove the presence of micropores (<2nm) with the isotherm you made?

Lines 258-259, 271: your values of cavity size etc. are very precise. How many measures did you make, what is the uncertainty?

Line 265: with the analysis you described at this state of the paper and the tables/figures you refer to; how can you judge on acidity?

Table 4: I do not understand why you use V and VBJH for the total pore volume. Please justify differences and impact on your material. I suppose that cutting at 3 nm and 10nm in the table is linked to a potential application not described here, but for the reader it's easier to look at the whole distribution than on a table.

Line 273 “near complete absence of micropores”: same question as for sample S2.

Figure 4: the adsorption and desorption branches are not superposed. Usually related to experimental issues.

Line 294-295: samples S1 and S2 are treated at 550°C. The first part of the paper relates to samples treated at lower temperatures (100°C for sample S1 and 350°C for sample S2) this thermal treatment will have an impact on crystallinity, and surface characteristics. How can you justify that the TPD-NH3 results are representative of samples S1 and S2? Why not having done all characterizations of S1 and S2 after a thermal treatment at 550°C?

Author Response

Dear Reviewer!

Thank you for your comments. We are very sorry for misprints in the text. We are glad to correct our mistakes and inaccuracies. All corrects are made and highlighted in yellow in the manuscript.

Point 1: Line 16 : alcoholates hydrolysis product: the usual term is “alcoxyde”, alcoholates is more generic.

Response 1: Thank you, the term ‘alkoxide’ was changed by the ‘alcoholate’ in the manuscript (17, 43, 146 lines).

Point 2: Line 81 : MS : MS as standing alone technique? do you mean ICP-MS?, ATG coupled MS?

Response 2: Mass spectrometry (MS) analysis was carried out at the heating of samples at the 10 °С/min in He current using a ThermoStar GSD 320 T gas analyzer (Pfeiffer Vacuum, USA) for the analysis of related gases resulting of samples heating. Yes, MS analysis is a standing technique for this purpose. ICP-MS was used for the element composition analysis.

Point 3: Line 97: what is Q in the equation?

Response 3: This reaction (102 line) is exothermic, i.e. it goes with the release of heat (Q – heat of the reaction). We removed Q from the equation to avoid any misunderstanding.

Point 4: Line 126: What is the standard “Barret-Joyner –Highland mechanism”? Usually BJH refers to the method of Barrett, Joyner and Halenda, a procedure for calculating pore size distributions from experimental isotherms using the Kelvin model of pore filling. Please describe this mechanism and add literature reference.

Response 4: We are so sorry for this terrible mistake. Of course, we meant Barrett-Joyner-Halenda mechanism. Corrections were made (130 line).

Point 5: Lines 247, 257, 258, 351: crystals? The term crystal is misused in the paper; with 100µm crystals you should have nice XRD patterns….

Response 5: Thank you for this comment. In order not to mislead the reader, the term ‘crystal’ is changed by a term ‘particle’ in all cases (244, 253, 255, 256, 263, 271, 357, 366 lines).

Point 6: Lines 253-254: “is a mesoporous compound with meso and micropores”: How can you prove the presence of micropores (<2nm) with the isotherm you made?

Response 6: Thank you for this comment. This sentence is not entirely true. "Micro" means pores with a diameter of less than 3 nm, which is not correct in this case. "Micro" was removed from the text of the manuscript (251 lines). The presence of micropores in the sample we judged by the distribution of pore volume by diameters (Table 1S in the Supplementary Material).

Point 7: Lines 258-259, 271: your values of cavity size etc. are very precise. How many measures did you make, what is the uncertainty?

Response 7: We determined the cavity sizes according to the SEM data. We selected a SEM image that most demonstrated the morphology of the sample, and determined the sizes of particles and cavities using a microscope.

Point 8: Line 265: with the analysis you described at this state of the paper and the tables/figures you refer to; how can you judge on acidity?

Response 8: We studied total acidity of surfaces of S3 sample and aluminas obtained by the heating of samples S1 and S2 using the Temperature-Programmed Desorption of Ammonia. To compare these samples and in the future to predict the acidic properties of the alumina surface derived from amorphous aluminum compounds obtaining by various methods, we show the curves for the desorption rate of ammonia on the desorption temperature and a table with data on the number of acid sites. Indeed, as we see from the figure 5 and the table 5 the amorphous alumina (S3) and the oxides derived from thermal decomposition of S1 and S2 have different acidity, which was shown in the conclusions.

Point 9: Table 4: I do not understand why you use V and VBJH for the total pore volume. Please justify differences and impact on your material. I suppose that cutting at 3 nm and 10nm in the table is linked to a potential application not described here, but for the reader it's easier to look at the whole distribution than on a table.

Response 9: V is total pore volume and was calculated by the last point of isotherm. VBJH was shown in the Table 4 for the pore volumes’ distributions over the pore diameters calculated from the curve of desorption isotherm by using the standard Barrett–Joyner–Halenda mechanism. The differences and calculating methods were described in research work [17] which was included in the References (125, 126, 130 lines).

Indeed, this distribution of pore volume with diameters, shown in Table 4, is necessary for further analysis of the catalytic properties. The whole distribution of pore volume of samples S2 and S3 was added to Supplementary materials in Table 1S (251-252, 273 lines).

Point 10: Line 273 “near complete absence of micropores”: same question as for sample S2.

Response 10: The absence of micropores was shown on the Table 1S in the Supplementary Material. ‘Near complete’ was removed from the manuscript (272 lines).

Point 11: Figure 4: the adsorption and desorption branches are not superposed. Usually related to experimental issues.

Response 11: The adsorption and desorption branches are not superposed due to capillary condensation effect, you know. However, isotherm of S2 sample have the greater divergence of branches probably because the sample possesses least a specific surface area among S2 and S3 samples.

Point 12: Line 294-295: samples S1 and S2 are treated at 550°C. The first part of the paper relates to samples treated at lower temperatures (100°C for sample S1 and 350°C for sample S2) this thermal treatment will have an impact on crystallinity, and surface characteristics. How can you justify that the TPD-NH3 results are representative of samples S1 and S2? Why not having done all characterizations of S1 and S2 after a thermal treatment at 550°C?

Response 12: To study the total acidity of the samples’ surface, it is necessary to remove hydroxyl groups from it. This is a standard procedure, aluminum compounds are heat treated at 550 °C. Of course, the crystallinity of these compounds will change, as shown in the article. Pseudoboehmite in S1 at this temperature forms gamma-alumina, which does not have an amorphous structure. The basic salts and amorphous aluminum hydroxide form amorphous alumina.

The fact is that the total acidity of these samples can be studied only after calcination at 550 °C, so we heated our samples. We investigated in manuscript the samples before heat treatment using a variety of methods due to the aim of the study was to compare amorphous aluminum compounds obtained by various popular methods, such as precipitation, hydrolysis of alcoholates and thermal decomposition of aluminum nitrate. In order to compare their acidic properties, which are a very important characteristic of the products obtained from them, the samples needed to be heat treat, which we did.

Best Regarding,

Kazan Federal University

PhD student

Aliya N. Mukhamed'yarova

Reviewer 2 Report

The paper reports comparative analysis of methods for producing amorphous aluminum compounds. The methods include precipitation and thermal decomposition. In addition, the physicochemical properties of the compounds are characterized and discussed, and a methodology for determining phase composition of amorphous aluminum compounds using a combination of characterization techniques is proposed. Unfortunately, the paper is difficult to read and comprehend, and apparently needs a comprehensive revision from an individual with a good understanding of English grammar rules. The errors either distract from the point being discussed or prevent a complete understanding of the discussion. I do not recommend acceptance of the paper in its current state. I encourage the authors to seriously consider the following points:

The paper is replete with grammatical mistakes that distract from the point(s) being made or muddle the point being made. Without a comprehensive revision, readers will find the paper difficult to read /understand. It does not seem enough thought is given to paragraphing; since paragraphs are the building blocks of effective writing, authors should consider reviewing the rules.

What are the enthalpies of the transformation reactions presented in the paper?

The precision of the data are inconsistent. What is the rationale for the difference in precision of the amount (wt%) from DSC and XRD?

For complete physicochemical characterization as claimed, won't a technique like XPS be required? – to probe the library of surface species and states.

Tables should also include the precursor type or synthesis method for easy comprehension. The use of S1 and S2 is not reader friendly. Alternatively, names that are coined from sample properties could be used to prevent readers from referring back to the experimental each time they have to compare samples.

Ambiguity should be avoided. For instance, instead of using words like "various methods," the specific methods should be simply stated.

BET description in the experimental section has a lot of redundant details. This method is well established and understood, thus the basis for the calculation of specific surface area is not necessary.

Author Response

Dear Reviewer!

Thank you for your comments. We are very sorry for misprints in the text. We are glad to correct our mistakes and inaccuracies. All corrects are made and highlighted in yellow in the manuscript.

Best Regards,

PhD student

Kazan Federal University, Russia

Aliya N. Mukhamedyarova

Point 1: The paper is replete with grammatical mistakes that distract from the point(s) being made or muddle the point being made. Without a comprehensive revision, readers will find the paper difficult to read /understand. It does not seem enough thought is given to paragraphing; since paragraphs are the building blocks of effective writing, authors should consider reviewing the rules.

Response 1: We are very sorry for grammatical mistakes and paragraphing. We tried to correct all them.

Point 2: What are the enthalpies of the transformation reactions presented in the paper?

Response 2: The water removal is recorded in the temperature range of the reaction. The enthalpy of a process obtained by the thermal analysis (TG/DSC) includes the sum of the enthalpy of the transformation reaction and the enthalpy of the water vaporization. The enthalpy of the water vaporization is also very difficult to calculate.

The enthalpy of this process:

is not calculated due to pseudoboehmite (Al2O3·nH2O) is identified on DSC curve as a shoulder.

The enthalpy of the I endothermic effect is -86.78 kJ/mol (for S1 sample) and -69.98 kJ/mol (for S2 sample).

The enthalpy of the II endothermic effect is -150.1 kJ/mol (for S1 sample) and -68.57 kJ/mol (for S2 sample).

The enthalpy of the endothermic effect of release of physical adsorption water from the sample S3 is -46.9 kJ/mol.

The enthalpy of the exothermic effect is 101.1 kJ/mol (for S1), 277.1 kJ/mol (for S2) and 66.6 kJ/mol (for S3).

Exothermic effect: Al2O3amorph γ-Al2O3 θ-Al2O3 α-Al2O3.

Point 3: The precision of the data are inconsistent. What is the rationale for the difference in precision of the amount (wt %) from DSC and XRD?

Response 3: Using XRPD it is possible to obtain information on the qualitative composition of the sample. Using the data of TG/DSC analysis, we calculated the quantitative composition of the synthesized amorphous aluminum compounds by weight loss of water and nitrogen oxide.

We are so sorry we do not understand the question.

Point 4: For complete physicochemical characterization as claimed, won't a technique like XPS be required? – to probe the library of surface species and states.

Response 4: Thank you for this comment. However, to compile physico-chemical characteristics the necessary data of the surface properties were given in the paper. Unfortunately, while we do not have the opportunity to analyze the sample using XPS analysis. In the future we will try to do it.

Point 5: Tables should also include the precursor type or synthesis method for easy comprehension. The use of S1 and S2 is not reader friendly. Alternatively, names that are coined from sample properties could be used to prevent readers from referring back to the experimental each time they have to compare samples.

Response 5: Thank you for this comment. Obtaining conditions were added in the tables.

Point 6: Ambiguity should be avoided. For instance, instead of using words like "various methods," the specific methods should be simply stated.

Response 6: Thank you, ‘various methods’ were changed in the whole text of paper.

Point 7: BET description in the experimental section has a lot of redundant details. This method is well established and understood, thus the basis for the calculation of specific surface area is not necessary.

Response 7: Superfluous information of calculating method was deleted (127 line).

Reviewer 3 Report

The manuscript is focused on the comparison of the microstructural and physicochemical properties of amorphous aluminium compounds obtained in three different ways. Interesting is the method for the quantitative calculation of the amorphous phase combing the XRD, TG/DSC and mass spectrometry.

The manuscript is well structured and supported by a complete analysis of the alumina samples also through nitrogen adsorption isotherm (surface area and nanoporosity) and SEM, with a detailed description of the results obtained in particular concerning the thermal analysis.

A suggestion is the substitution of the SEM images with images at the same magnification for each sample.

In the opinion of this referee the manuscript is suitable for the publication in "Coating".

Author Response

Dear Reviewer!

Thank you so much for your positive comments. We are glad to correct our mistakes and inaccuracies. All corrects are made and highlighted in yellow in the manuscript.

Best Regards,

PhD student

Kazan Federa University, Russia

Aliya N. Mukhamedyarova

Point 1: A suggestion is the substitution of the SEM images with images at the same magnification for each sample.

Response 1: Thank you for the comment. SEM images at the same magnification were added in the manuscript.

Round 2

Reviewer 1 Report

Point 1: Line 16 : alcoholates hydrolysis product: the usual term is “alcoxyde”, alcoholates is more generic.

My point was to use alcoxyde rather than alcoholate

Point 7: Lines 258-259, 271: your values of cavity size etc. are very precise. How many measures did you make, what is the uncertainty?

you did not respond to this point: it is clear that you use an SEM and make measures. However your values are very precise: for example, line 244: 46-846nm. How can you justify this precision? If it's just an order of magnitude, something like "50-850nm" would be expected. Please justify.

Point 9:  I still do not understand why you need both in THIS paper. Also, I still think that the plot of the distribution, rather than table 4, helps understanding the narrow disptribution you speak of. I had to copy your supplementary material into Excel and plot it to be able to understand what you are trying to say.

Point 12:  The point is that after heat treatment you have a completely different material and that the acidity you get is in no point related to the structural investigation you made before. There is a lack of consistancy between part 1 and part 2 of your work.

Author Response

We want to thank you for your careful review of our paper, and for the comments and suggestions concerning the research work. All of them will be taken into account during a major revision of the paper. And we believe it will be significantly improved during the process.

Reviewer 2 Report

The revision has not addressed the many grammatical mistakes.

Manuscript still requires editing.
Author Response

 Dear Reviewer 2!

We want to thank you for your careful review of our paper, and for the comments and suggestions concerning the research work. All of them will be taken into account during a major revision of the paper. And we believe it will be significantly improved during the process.

Best Regards,

PhD student

Aliya N. Mukhamed'yarova

This manuscript is a resubmission of an earlier submission. The following is a list of the peer review reports and author responses from that submission.

Round 1

Reviewer 1 Report

The manuscript describes three methods to synthesize amorphous Aluminum compounds.

The overall manuscript needs improvement and I do not recommend its publication in coatings in the present form.

The aim of the study is not clearly defined: shape of the material is with no link to coatings, and if there is an indirect link, through a specific coating application, it does not appear in the manuscript.

Moreover, I suggest a reorganization of the manuscript away from a separate descriptions of the results and the discussions to results & discussions, were the experimental results are commented directly, avoiding repetition.

The methods need improvement, which is necessary to support the results and discussions.

The purity of the raw materials used is not given. Therefore, the discussion about impurities between samples lacks interest because it can be changed by using reactants with different purities. Comparing with reactants purities will allow to point specifically on impact of the synthesis method.

The synthesis method of S3 is incomplete. The synthesis method of the “non crystalline aluminum hydroxide” needs to be described. How can you be sure that you have a hydroxide? Aluminum alcoxides in aqueous media are extremely reactive and will form Al-O-Al bonds.

The morphologies obtained are directly linked to the synthesis method. The work should be completed with SEM images of the starting material: The cracks seen in sample S1 are dependent of the drying (by the way, the drying is not described in the methods part). S2 should be linked to the initial crystal size and S3 is due to the synthesis method of the “non crystalline aluminum hydroxide”. The linear pores that are observed can be due to immiscibility of alcoxide and the solvent (what solvent is used?), in this case they are already present before thermal treatment. In Table 3, the differences between V, VBJH are not clear. If you could not obtain an isotherm, where does the VBJH comes from? I would recommend plotting the isotherms rather than the table, as “narrow pore size distribution” makes only sense if you have the whole distribution.

The TPD-NH3 methodology is not sufficiently described. In the manuscript you state that a thermal treatment was used for S1 and S2 but not S3. You do not mention any treatment to remove water adsorption prior to the measurements. S3 has a very high specific surface area and water adsorption will occur naturally in air. What kind of detector did you use? Indeed, it is very difficult to separate water and NH3. I suspect that part of your signal is due to water, impacting interpretation.

You can improve the paper by insisting on the aim of your study. What are the characteristics you are looking for? Are you looking to avoid the crystallization? If the aim is to develop a methodology to quantify the amorphous versus pseudo-boehmite part, say it clearly in abstract and conclusions., and rewrite your paper accordingly.

Additional comments:

Avoid acronyms in the abstract and in the text. iCAP RQ is a specific equipment to make ICP-MS. Checking back in the methods section, you used a iCAPQc, which one did you use? What is GOST (line 199)?

Line 30: check spelling of boxite, à bauxite

Line 144: “crystals of ~100µm” the terms crystal is misused here.